ⓐ | **Open Peer Review** | Epidemiology | Research Article

# The effect of changing diagnostic method from culture to PCR on the number of episodes of human campylobacteriosis in Denmark: a retrospective study (2015–2022)

Guido Benedetti,[1] Christian Holm Hansen,[1] Anna Tølbøll Svendsen,[1,2,3] Katrine Grimstrup Joensen,[4] Gitte Sørensen,[4] Anne Line Engsbro,[5,6] Mia Torpdahl,[4] Eva Møller Nielsen,[4] Steen Ethelberg[1,7]

**ABSTRACT**   We investigated whether the introduction of polymerase chain reaction (PCR) to replace culture as the primary diagnostic method for *Campylobacter* species in the Danish Departments of Clinical Microbiology (DCMs) was associated with a systematic change in the number of reported episodes of human campylobacteriosis from 2015 to 2022. We described a hypothetical scenario where PCR was solely used from 2018 to 2021. We analyzed the effect of primary diagnostic methods (culture or PCR) on the number of weekly episodes of human campylobacteriosis in a negative binomial regression adjusting for time, seasonality, COVID-19 restrictions, and DCMs. Furthermore, we applied the estimated PCR effect to the number of episodes that was identified using culture. Overall, PCR was associated with a 43% estimated increase of weekly campylobacteriosis episodes [rate ratio (RR) 1.43, 95% confidence interval (95% CI) 1.34–1.53]. Up to 17%–25% additional episodes would have been reported yearly from 2018 to 2021, had PCR been solely used. Through the lens of laboratory diagnostic methods, we described a systematic change in the number of reported episodes of human campylobacteriosis in Denmark. This is operationally relevant as considerably more episodes would have been identified had PCR been solely used. Changes in diagnostic methods should be considered in the epidemiological analysis of historical data.

**IMPORTANCE**   This study is important because it shows the potential epidemiological silence associated with the use of culture as the primary diagnostic method for the laboratory identification of human campylobacteriosis. Also, we show how polymerase chain reaction methods are associated with a systematic increase in the number of human campylobacteriosis episodes as reported by routine disease surveillance. These findings are operationally relevant and have public health implications because they tell how crucial it is to consider changes in diagnostic methods, e.g., in the epidemiological analysis of historical data and in the interpretation of future data in light of the past. We also believe that this study highlights how the synergy between microbiology and epidemiology is essential for disease surveillance.

**KEYWORDS**   foodborne diseases, epidemiological monitoring, public health surveillance, *Campylobacter*, culture techniques, polymerase chain reaction

Infections with bacteria of *Campylobacter* species are the most common reason for human gastroenteritis among the countries of the European Union (EU) and the European Economic Area (EEA) (1). *Campylobacter* species are largely present in warm-blooded animals, and food and the environment are primary sources of infection for humans. Despite being commonly mild, campylobacteriosis (an infection with

Address correspondence to Guido Benedetti, gube@ssi.dk.

The authors declare no conflict of interest.

*Campylobacter* species) can be particularly severe in childhood, old age, and for fragile groups like the immunosuppressed (2, 3). In 2021, there were almost 130,000 confirmed cases of human campylobacteriosis across 30 EU/EEA countries, even while several EU member states do not have complete national coverage in their surveillance systems, e.g., France, Italy, the Netherlands, and Spain (1).

In 2022, Denmark registered 5,142 laboratory-confirmed episodes of campylobacteriosis (*Campylobacter jejuni* and *Campylobacter coli*) corresponding to an incidence of 87 episodes per 100,000 inhabitants (4). The average annual number of registered episodes from 2015 to 2022 was 4,482 (5). A drop of travel-related episodes was observed in conjunction with the SARS-CoV-2 pandemic in 2020 and 2021 (6).

In Denmark, the diagnosis of infectious diseases in primary care and hospitals relies on the nationwide network of Departments of Clinical Microbiology (DCMs). The collaboration between DCMs and clinical settings "enables rapid and relevant testing of patient samples and optimal antimicrobial treatment of the individual patient" (7). For many years, stool culture methods have been the first choice for the laboratory identification of human campylobacteriosis in the Danish DCM. Since 2012, DCMs have at different times begun replacing culture with polymerase chain reaction (PCR) directly on fecal matter as their primary diagnostic method.

For the identification of *Campylobacter* species in humans, culture-based methods are reported to have lower sensitivity (8, 9) and to produce "substantial underdetection" (10) in comparison to PCR methods. However, the hypothesis that shifting from culture to PCR methods might have affected the number of episodes of human campylobacteriosis that is reported by the Danish national routine surveillance has not been explored yet, while the historical number of episodes might have been underreported due to culture methods.

Through analysis of data recorded in the nationwide Danish microbiological surveillance system, this study aimed to determine whether the introduction of PCR as the primary diagnostic method for human campylobacteriosis in the Danish DCMs was associated with a systematic change in the number of reported episodes from January

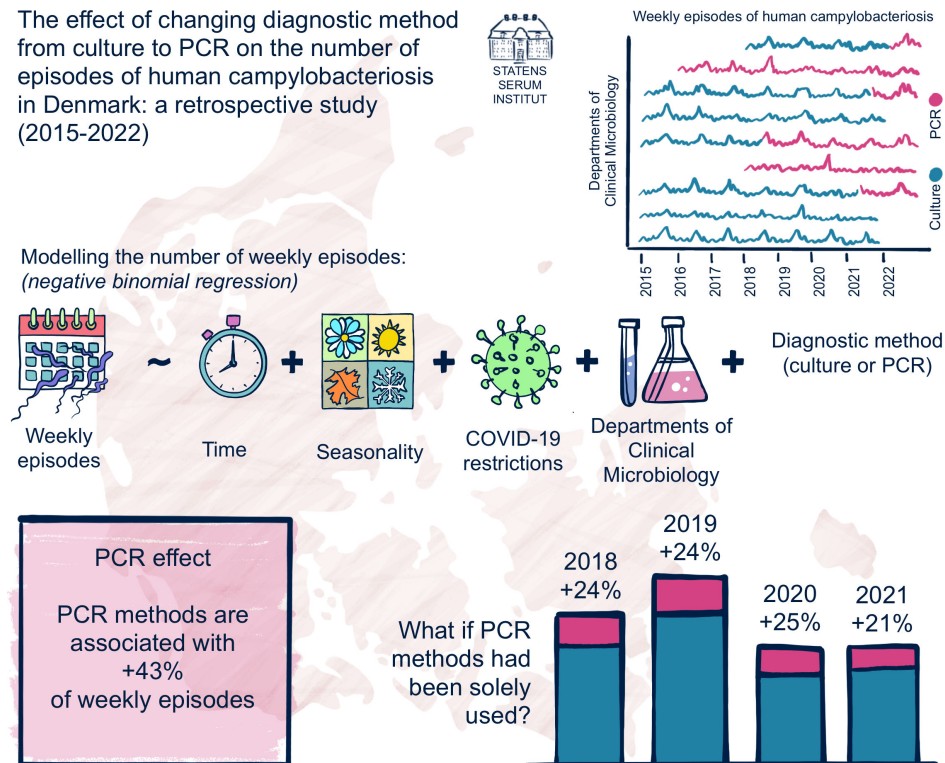

**FIG 1** Overview of the study.

2015 to December 2022 and to estimate the magnitude of this effect (henceforth called "PCR effect"). Second, the study aimed to estimate the number of episodes of human campylobacteriosis that would have been reported if PCR methods had been solely used across all DCMs from January 2018 to December 2021. Figure 1 gives an overview of this study.

## MATERIALS AND METHODS

### Study design

This was a retrospective data analysis study of routinely collected surveillance data investigating the number of reported episodes of human campylobacteriosis in Denmark from January 2015 to December 2022.

### The Danish DCMs

In Denmark, clinicians send human samples for diagnostic testing to 1 of 10 DCMs, and all test results are automatically submitted to the Danish Microbiology Database (MiBa) (6, 11). The testing setup varies between DCMs, and in 2018, a nationwide survey was conducted on the diagnostic methods used for the detection of enteric bacteria, virus, and parasites. The survey described whether the primary method for *Campylobacter* species was culture or PCR, and whether a change (from culture to PCR) had occurred and when (month and year). No further information about the utilized laboratory methods was available in the survey. The survey was updated in 2021, and further, routine updates are planned (12, 13).

Regardless of the primary diagnostic method, *Campylobacter* species have been routinely investigated in every sample that clinicians submit to DCMs for the testing of gastrointestinal bacterial pathogens. Over time, there has been no standardization of culture methods across DCMs, and as they changed their primary diagnostic method from culture to PCR, they introduced a variety of in-house PCR protocols or commercial PCR kits. Therefore, also the PCR setup has not been standardized across DCMs. When describing a PCR effect, this study described the effect of a combination of diverse PCR protocols and kits.

### Data sources

In Denmark, findings of *Campylobacter* species are notifiable, and patient episodes are registered in the National Register of Enteric Pathogens (abbreviated "TBR," in Danish), which is maintained by the national reference laboratory at Statens Serum Institut (SSI). Within the study period, TBR was based on manual reports from the DCMs until 2017. Since 2018, it has been based on information from MiBa. The TBR case definition for human infections with *Campylobacter* species is event based with a time window of 6 months, i.e., "each patient-infectious agent (species) combination is only recorded once in any 6-month period," which defined an episode in this study (4). Information to link episodes of campylobacteriosis and DCMs in TBR is available starting in 2015.

Denmark introduced a series of nonpharmaceutical interventions in response to the COVID-19 pandemic (14), and we assumed that these reduced the exposure to *Campylobacter* species from domestic sources and travels abroad, that they changed the health-care seeking behavior at primary care and, particularly in its initial phase, the DCMs' test capacity. To account for this, information about the strictness of lockdown policies due to the COVID-19 pandemic in Denmark was retrieved as the "stringency index" (defined as an index of the "containment and closure policies, sometimes referred to as lockdown policies") from the Oxford COVID-19 Government Response Tracker, Blavatnik School of Government, University of Oxford (CC BY License) on 02 March 2023 (15).

## Inclusion criteria and data preparation

Nine DCMs were included in this analysis: DCMs Herlev, Hvidovre, Lillebælt, Odense, Slagelse, Sydvestjysk, Sønderjylland, Aalborg, and Aarhus. DCM Rigshospitalet was excluded because it serves the needs of patients admitted to tertiary care, and it does not receive samples from primary care. All episodes of human campylobacteriosis available in TBR during the study period were eligible for inclusion regardless of the patient's age, sex, and travel history or whether they pertained to periods with known outbreaks of human campylobacteriosis.

We excluded episodes with missing information about the related DCM. We excluded information from the Danish national reference laboratory at SSI, which is also available in TBR. We excluded data from two DCMs before 2018 (DCM-A, DCM-B in Fig. 2) because their observations could not be distinguished, and they were known to use different primary diagnostic methods (one using culture and the other PCR). We excluded data from one DCM before 2016 (DCM-C in Fig. 2) because no episode of human campylo-bacteriosis was reported before that date. We excluded data from three DCMs in 2022 (DCM-F, DCM-G, DCM-H in Fig. 2) because their primary diagnostic methods were not documented for that year at the time of this analysis. The remaining episodes of human campylobacteriosis comprised the data set of this analysis (henceforth called the "base data set").

Data were summarized by DCM and epidemiological week. Observations from occurring week 53 were excluded to ensure the same number of weeks (52) across years. For each DCM, we assigned a binary value to every week of observation to define when the primary diagnostic method was culture or PCR. We used a weekly average of the daily COVID-19 stringency index for Denmark.

## Statistical analysis

Analyses were performed with the language and environment R (16). The number of weekly episodes of human campylobacteriosis was analyzed in a negative binomial regression with adjustment for time (weeks), seasonality (given the seasonal pattern of campylobacteriosis in Denmark) (17), the restrictions of the SARS-CoV-2 pandemic ("stringency index"), the DCMs that performed the laboratory analysis, and the primary diagnostic method at DCM level. The general formula of the model was the same as described by Byers et al. (18). The model (henceforth called "base model") was run with the "glm.nb" function of the R package "MASS" (19). Table 1 describes the explanatory variables of the base model. The outcome variable was the modeled count of new weekly episodes of human campylobacteriosis at DCM level.

The exponential of the estimated coefficient (rate ratio: RR) for the explanatory variable "primary diagnostic method" and its 95% confidence interval (95% CI) measured the PCR effect in the base model, and it can be interpreted as the weekly percentage change in the number of episodes associated with using PCR compared to culture methods.

**TABLE 1** Explanatory variables of the base model

| Explanatory variable | Description |
|---|---|
| Time | Numeric, integer. Range: 1–416, number of weeks from week 1, 2015 to week 52, 2022 (excluding week 53) |
| Seasonality | Numeric, continuous. Four variables as the sinus and cosinus of the variable time (in weeks) over periods of 52 and 26 weeks and measured from week 1, 2015 |
| Restrictions due to the SARS-CoV-2 pandemic | Numeric, integer. Range 0–100. Weekly average of the daily "stringency index" for Denmark |
| Departments of Clinical Microbiology | Categorical, nominal. Levels: DCM-A, DCM-B, DCM-C, DCM-D, DCM-E, DCM-F, DCM-G, DCM-H, DCM-I |
| Primary diagnostic method | Categorical, dichotomous. Levels: 0 = culture, 1 = PCR |

## Model diagnostics

We measured: the median deviance of the residuals of the base model, the overdispersion of the base model as the ratio of its residual deviance and the degrees of freedom, the collinearity of explanatory variables as the variance inflation factor of the model (with the function "mctest" of the R package "mctest") (20), the autocorrelation of the model residuals, by estimating its autocorrelation function (with the function "acf" of the R package "stats") (21), and by testing the autocorrelation of disturbances via the Durbin-Watson test (with the function "dwtest" of the R package "lmtest") (22). We designed two approaches to address autocorrelation of residuals: (i) running the base model on a data set of size equal to that of the base data set and obtained by its random sampling with replacement; (ii) running the base model on 1,000 random draws of 806 weekly observations each (a quarter of the weekly observations in the base data set) in a Monte Carlo simulation and empirically measuring the PCR effect as the mean estimate of the simulation and its 95% CI as the 2.5th and 97.5th percentiles of the estimates of the simulation.

## Sensitivity analysis

The base model was also run on the base data set with an interaction term between the explanatory variables DCM and the primary diagnostic methods, thereby estimating separate PCR effects by each DCM (model-2). Supplementary materials further describe the sensitivity analysis.

## Application of the PCR effect

To estimate the number of campylobacteriosis episodes in Denmark that would have been detected if PCR methods had been solely used across all DCMs, we applied the estimated PCR effects (from the base model and model-2) to the observed yearly number of episodes of human campylobacteriosis, i.e., multiplying the reported episodes (analyzed with culture methods) by the measured PCR effects. This analysis was restricted to the period from 2018 to 2021, when TBR was based on information from MiBa, and data were complete for all DCMs.

## RESULTS

All 35,984 episodes of human campylobacteriosis available in TBR from January 2015 to December 2022 were retrieved. After removing episodes from DCM Rigshospitalet (302), episodes missing information about the related DCMs (1,142), episodes from the Danish national reference laboratory at SSI (92), and episodes from week 53 (110), 34,338 episodes were included for analysis.

Figure 2 describes the weekly number of episodes of human campylobacteriosis in TBR from January 2015 to December 2022 that were included in the analysis by DCM and their primary diagnostic method (culture or PCR).

During the study period, four of nine DCMs changed their primary diagnostic method from culture to PCR (DCM-A, DCM-D, DCM-E, DCM-I), while two had previously introduced PCR (DCM-B, DCM-C). The other three DCMs had culture as the primary diagnostic method during the study period (DCM-F, DCM-G, DCM-H).

The base model associated +43% weekly *Campylobacter* episodes when PCR was the primary diagnostic method at DCM level (RR 1.43, 95% CI 1.34–1.53). Estimates of the remaining regression coefficients are available as Supplementary Materials. The median deviance of the model residuals was −0.1 [interquartile range −0.8–0.5]. The ratio of the residual deviance and the degrees of freedom was 1.06. No collinearity was detected in the base model. The residuals of the base model showed autocorrelation up to a 9-week lag (Durbin-Watson statistic = 1.72, $P$-value <0.01).

Both approaches addressing autocorrelation of residuals yielded similar PCR effects to the base model. When the base model was run on a data set of size equal to that of the base data set and obtained by its random sampling with replacement, residuals

**TABLE 2** Estimated number (and relative increase) of episodes of human campylobacteriosis in Denmark from January 2018 to December 2021 in a hypothetical scenario where only PCR methods were used

|  | 2018 | 2019 | 2020 | 2021 |
|---|---|---|---|---|
| Number of DCMs using PCR at the end of the year, out of nine DCMs | 3/9 | 3/9 | 3/9 | 5/9 |
| Analyzed episodes[a] | 4,482 | 5,324 | 3,680 | 3,645 |
| PCR effect applied (base model) | 5,418 | 6,388 | 4,425 | 4,248 |
| *Relative increase* | *+21%* | *+20%* | *+20%* | *+17%* |
| DCM-specific PCR effect applied (model-2)[b] | 5,571 | 6,616 | 4,608 | 4,412 |
| *Relative increase* | *+24%* | *+24%* | *+25%* | *+21%* |

[a]The number of episodes here reported does not correspond to the yearly number of registered episodes of campylobacteriosis in Denmark.
[b]This model measured DCM-specific PCR effects. The PCR effect of the base model was applied to the weekly observations from DCMs that did not introduce PCR methods during the entire study period.

were independent after a 1-week lag (Durbin-Watson statistic = 2.02, *P*-value <0.68). This model associated +40% weekly episodes when PCR methods were in place at DCM level (RR 1.40, 95% CI 1.30–1.50). The Monte Carlo simulation running the base model on 1,000 random draws from the base data set resulted in a mean RR of the PCR effect equal to 1.43 (empirically derived 95% CI 1.25–1.62, Supplementary Materials).

All models that were run as part of the sensitivity analysis measured similar PCR effects to that of the base model, and all suggested an increased number of episodes associated with PCR methods (Supplementary Materials).

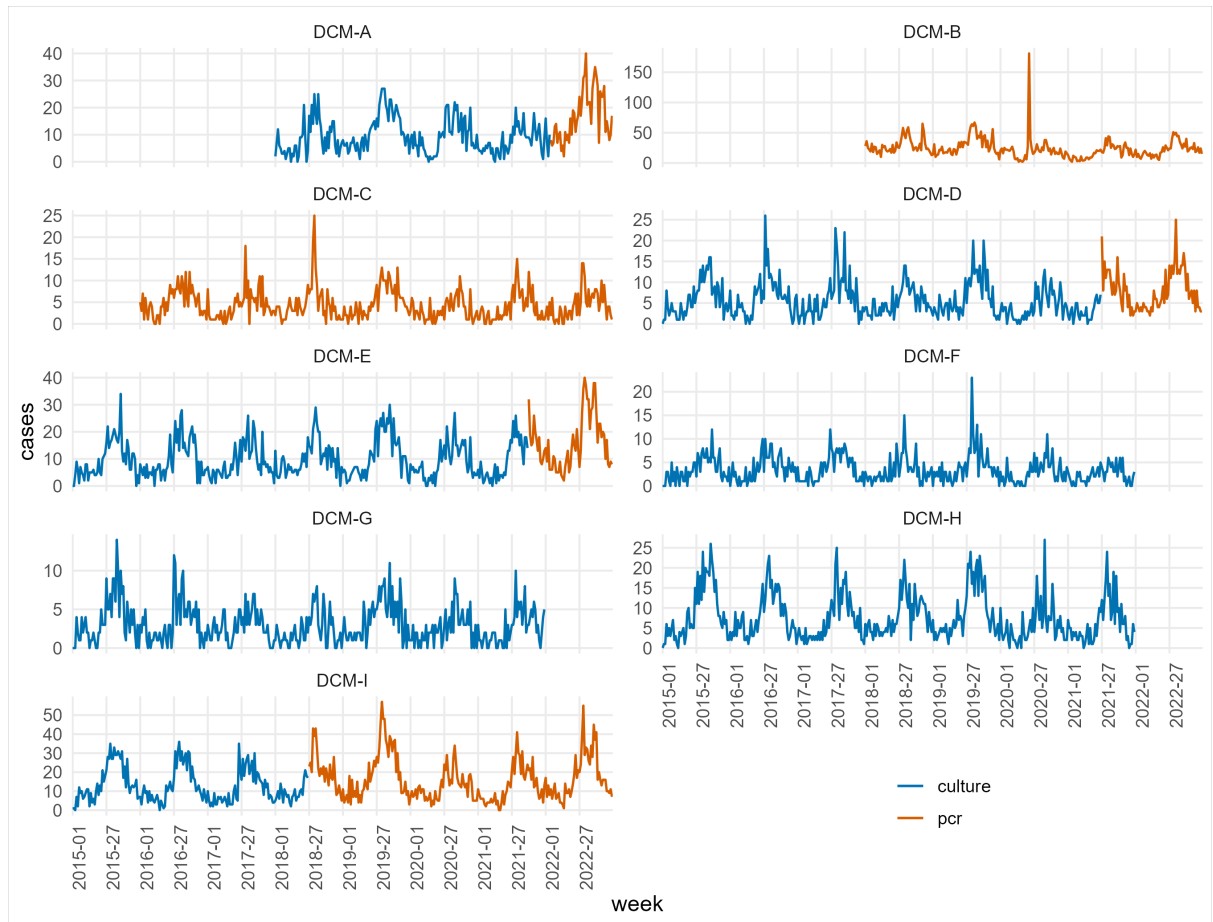

**FIG 2** Weekly number of episodes of human campylobacteriosis available in TBR from January 2015 to December 2022 that were included in the analysis (base data set) by DCMs and their primary diagnostic method (culture or PCR).

Table 2 shows the estimated yearly number (and their relative increase) of episodes of human campylobacteriosis in Denmark from January 2018 to December 2021 in a hypothetical scenario where PCR methods had been solely used. The yearly number of reported episodes would have been 20%–25% higher in the period 2018–2020 (when three of nine DCMs used PCR methods) and 17%–21% higher in 2021 (when five of nine DCMs used PCR methods).

## DISCUSSION

We quantified the difference that episodes associated with PCR methods made to the surveillance of human campylobacteriosis in Denmark from 2015 to 2022. The existence of an effect was robust to different model choices, and it allowed us to estimate the occurrence of human campylobacteriosis in a hypothetical scenario where PCR methods had been solely used. To our knowledge, this was the first study to quantify the difference between culture and PCR methods in the number of reported episodes of human campylobacteriosis by routine surveillance. These findings are relevant because they allow a better understanding of the historical trends of human campylobacteriosis and a more thorough interpretation of future epidemiological data in light of the past. Additionally, it is important to consider that Denmark, like other countries, has implemented a series of action plans, aiming to reduce gastrointestinal infections through various interventions and collaborations across the public health, animal health, and food safety sectors. The Danish action plans have been evaluated by measuring if a reduction in the number of human cases was obtained; thus, when interpreting the impact of such action plans, it is also paramount to consider the effect different laboratory methods have on the detection of *Campylobacter* species.

The respective operational advantages of culture and PCR methods in the laboratory diagnosis of gastrointestinal infections have been already thoroughly explored. Generally, PCR methods are associated with better detection rates, while traditional culture methods remain fundamental to yield isolates for genomic sequencing (23–29). However, this was not a controlled, laboratory study aiming to explore and measure differences in the detection rates of culture and PCR methods. This was a retrospective study making use of real-life, routinely collected surveillance data and measuring the difference that accounting for variations in diagnostic methods can make to the surveillance of human campylobacteriosis when handling data from different laboratories or time periods.

Using historical data for the surveillance of infectious diseases requires the study of several factors (30), such as time trends and seasonality (31), previous epidemic or outbreak periods (32, 33), changes in case definitions and reporting methods (34, 35), data representativeness and completeness (36, 37), and even laboratory practices (38). Understanding these factors is fundamental, for example, to design alert systems and to set thresholds for extreme observations based on historical data. The difference that episodes associated with PCR methods made to the surveillance of human campylobacteriosis in Denmark (i.e., +43% weekly episodes) and the substantial correspondence of all the explored models confirm the operational challenge that changes in the laboratory diagnostic methods can represent to the routine surveillance of campylobacteriosis. Furthermore, the estimated yearly number of episodes of human campylobacteriosis in a hypothetical scenario where PCR methods had been solely used gave a quantitative example of the potential epidemiological silence associated with culture methods. However, it is important to highlight here that the reported increases (+20%–25% in 2018–2020 and +17%–21% in 2021) are not generalizable because they depend on the relative number of DCMs using culture or PCR.

From 2018 to 2022, Denmark reported crude annual incidence rates of human campylobacteriosis (*Campylobacter jejuni* and *Campylobacter coli*) of 79, 93, 64, 64, and 87 episodes per 100,000 inhabitants, respectively (4, 6, 39–42). However, despite the increased number of episodes associated with PCR methods and despite the increased population being covered by DCMs implementing PCR methods, the reported incidence

data do not appear to reflect an increase over time. Possible explanations for this include that the SARS-CoV-2 pandemic accounted for a drop in the number of reported travel-related episodes (6), while the higher incidence in 2019 has been related to a large foodborne outbreak (40).

We recommend accounting for the known, historical variations of laboratory diagnostic methods when conducting routine surveillance of campylobacteriosis. Furthermore, we recommend exploring the existence of similar effects for other pathogens of gastrointestinal relevance.

## Limitations

The explanatory variables included in the analysis were possibly not stable throughout the entire study period because of the change operated in 2018, when MiBa replaced the notification reports that DCMs sent to SSI as the source of TBR. Additionally, it was not possible to retrieve a complete time series for episodes of campylobacteriosis for each DCM from 2015 to 2022. Only four DCMs changed their primary diagnostic method for *Campylobacter* species from culture to PCR during the study period. Moreover, changes occurred after the second half of 2021 for three of them, making the time with PCR as the primary diagnostic method limited to appraise the effect of such a change. However, the similarities between the PCR effect measured using the base model and those measured in the sensitivity analysis suggest the appropriateness of our design.

TBR reports episodes of infections with enteric pathogens, but it does not provide information about the samples that were taken and the tests that were performed. Currently, we have no knowledge of any historical or geographical variation of the number of taken samples and performed tests that is relevant to human campylobacteriosis, but they could have affected our results by influencing the number of identified episodes.

From a similar perspective, it is worth noticing that there is no evidence of changes in the population living in Denmark (e.g., in its size or in its dietary habits) that might have occurred during the study period and that could contribute explaining our results alongside PCR methods. The registered population living in Denmark increased from 5.7 million in 2015 to 5.9 million in 2022, with constant growth rates and no evident regional variations (43, 44). The self-reported dietary habits of the population during our study period also do not offer a plausible explanation for the increased number of registered episodes of human campylobacteriosis with PCR methods: 18% of the population in 2017 and 15% in 2021 declared healthy dietary habits in the Danish National Health Survey by the Danish Health Authority (45), and 1.8% of the population in 2017 and 3% in 2022 had a vegetarian diet (46, 47).

PCR methods were introduced at different times and in a share of the Danish DCMs only (up to six out of nine DCMs at the time of this analysis). Any alternative explanation to the increased number of episodes should fit a similar pattern in time and place. Additionally, our models included time as an explanatory variable (calendar effect), which contributed capturing changes in the population's characteristics.

This study was possible, thanks to the commitment of the Danish DCMs, which informed the nationwide survey on the diagnostic methods used for the detection of enteric bacteria, virus, and parasites (12, 13). Over time, DCMs independently defined their analysis methods (both for culture and PCR), which were therefore not standardized. Unfortunately, the aforementioned survey did not provide further details about the different protocols, kits, or materials that were utilized. Theoretically, the PCR effect measured in our study might have reflected site-specific variations, but the results that were yielded by the sensitivity analysis did not support this possibility, and DCMs showed consistent PCR effects. Particularly, it was unknown if and what *Campylobacter* species other than *Campylobacter jejuni* and *Campylobacter coli*—the most commonly associated species with the disease in humans (2)—were targeted by the different DCMs with PCR methods. Since the subtyping of human isolates by whole-genome sequencing

was introduced in Denmark in 2019, *Campylobacter jejuni* and *Campylobacter coli* have constantly remained the most identified species (4, 48).

Finally, the aim of our study could have been broadened by including other pathogens of gastrointestinal relevance alongside *Campylobacter* species. However, the comparison of different pathogen-specific PCR effects would have introduced several methodological limitations. The incidence of campylobacteriosis (*Campylobacter jejuni* and *Campylobacter coli*), all *Salmonella* species, and Shiga toxin-producing *Escherichia coli* was equal to 87, 15, and 23 episodes per 100,000 inhabitants in 2022 in Denmark, respectively (4). Handling data of such a different magnitude would have necessarily increased the noise and uncertainty of the PCR-effect estimates. As much as for *Campylobacter* species, the methods to measure the effect of laboratory changes on the surveillance of other gastrointestinal pathogens should be tailored to their different epidemiological distributions in the population.

## Conclusion

The synergy between epidemiology and microbiology is essential for disease surveillance. Through the lens of laboratory diagnostic methods, we described a change in the number of reported episodes of human campylobacteriosis in Denmark, which is of operational relevance. Changes in diagnostic methods should be considered in the epidemiological analysis of historical data and when evaluating interventions.

### ACKNOWLEDGMENTS

We thank all the colleagues from the Danish Departments of Clinical Microbiology and the working group that collected information about the diagnostic methods for the detection of enteric bacteria, virus, and parasites.

Special thanks to Caroline Eves for the support provided in the editorial phase of this manuscript.

The Danish National Health Survey was funded by The Capital Region, Region Zealand, The South Denmark Region, The Central Denmark Region, The North Denmark Region, The Ministry of Health and the National Institute of Public Health, University of Southern Denmark.

This research did not receive any specific grant from funding agencies in the public, commercial, or not-for-profit sectors.

G.B., C.H.H., and S.E. substantially contributed to the conception and design of the work. G.B., C.H.H., A.L.E., E.M.N., and S.E. substantially contributed to the analysis. G.B., C.H.H., A.T.S., K.G.J., G.S., A.L.E., M.T., E.M.N., and S.E. substantially contributed to the interpretation of data for the work. G.B. drafted the work. C.H.H., A.T.S., K.G.J., G.S., A.L.E., M.T., E.M.N., and S.E. critically revised the work for important intellectual content. All authors approved the final version of the manuscript to be published. All authors agree to be accountable for all aspects of the work in ensuring that questions related to the accuracy or integrity of any part of the work are appropriately investigated and resolved.

The authors declare that they have no known competing financial interests or personal relationships that could have appeared to influence the work reported in this paper.

### AUTHOR AFFILIATIONS

[1]Department of Infectious Disease Epidemiology and Prevention, Statens Serum Institut, Copenhagen, Denmark
[2]Department of Medicine, Zealand University Hospital, Køge, Denmark
[3]Department of Clinical Medicine, University of Copenhagen, Copenhagen, Denmark
[4]Department of Bacteria, Parasites and Fungi, Statens Serum Institut, Copenhagen, Denmark
[5]Department of Clinical Microbiology, Zealand University Hospital, Slagelse, Denmark

[6]Hvidovre University Hospital, Hvidovre, Denmark

[7]Department of Public Health, Global Health Section, University of Copenhagen, Copenhagen, Denmark

## AUTHOR ORCIDs

Guido Benedetti  http://orcid.org/0000-0002-0788-8562

## DATA AVAILABILITY

Because of patient confidentiality, the data from the Danish National Register of Enteric Pathogens (abbreviated "TBR", in Danish) that were analyzed in this study are not in the public domain. Researchers are invited to contact the authors of this study for any related enquiry. This study utilized publicly available data from the Oxford COVID-19 Government Response Tracker, Blavatnik School of Government, University of Oxford (CC BY License) – accessed on 02 March 2023 (15).

## ETHICS APPROVAL

According to Danish law, ethical approval is not required for studies utilizing anonymized, aggregated, and register-based data. This study was performed as a national disease surveillance activity under the auspices of the Statens Serum Institut as per the Danish Health Act § 222 (49).

## ADDITIONAL FILES

The following material is available online.

### Supplemental Material

**Supplemental material (Spectrum03418-23-s0001.pdf).** Supplemental results: summary statistics of the base model, autocorrelation of residuals, and sensitivity analysis.

### Open Peer Review

**PEER REVIEW HISTORY (review-history.pdf).** An accounting of the reviewer comments and feedback.

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
