## [Reviewer comments · Microbiology Spectrum]

Microbiology Spectrum

The effect of changing diagnostic method from culture to PCR on the number of episodes of human campylobacteriosis in Denmark: a retrospective study (2015-2022)

Guido Benedetti, Christian Holm Hansen, Anna Tølbøll Svendsen, Katrine Grimstrup Joensen, Gitte Sørensen, Anne Engsbro, Mia Torpdahl, Eva Møller Nielsen, and Steen Ethelberg

Corresponding Author(s): Guido Benedetti, Statens Serum Institut

Review Timeline:

Submission Date:

September 22, 2023

Accepted:

November 13, 2023

Editor: Jasna Kovac

Reviewer(s): The reviewers have opted to remain anonymous.

Transaction Report:

DOI: <https://doi.org/10.1128/spectrum.03418-23>

Re: Spectrum03418-23 (The effect of changing diagnostic method from culture to PCR on the number of episodes of human campylobacteriosis in Denmark: a retrospective study (2015-2022))

Dear Dr. Guido Benedetti:

Your manuscript has been accepted, and I am forwarding it to the ASM production staff for publication. Your paper will first be checked to make sure all elements meet the technical requirements. ASM staff will contact you if anything needs to be revised before copyediting and production can begin. Otherwise, you will be notified when your proofs are ready to be viewed.

Sincerely,
Jasna Kovac
Editor
Microbiology Spectrum